# A Multimodal Multi-Stage Deep Learning Model for the Diagnosis of Alzheimer’s Disease Using EEG Measurements

**DOI:** 10.3390/neurolint17060091

**Published:** 2025-06-13

**Authors:** Tuan Vo, Ali K. Ibrahim, Hanqi Zhuang

**Affiliations:** EECS Department, Florida Atlantic University, Boca Raton, FL 33431, USA; aibrahim2014@fau.edu (A.K.I.); zhuang@fau.edu (H.Z.)

**Keywords:** dementia, Alzheimer’s disease, frontotemporal dementia, deep learning, spectrogram, scalogram, Hilbert spectrum, convolutional neural network, multimodality

## Abstract

**Background/Objectives:** Alzheimer’s disease (AD) is a progressively debilitating neurodegenerative disorder characterized by the accumulation of abnormal proteins, such as amyloid-beta plaques and tau tangles, leading to disruptions in memory storage and neuronal degeneration. Despite its portability, non-invasiveness, and cost-effectiveness, electroencephalography (EEG) as a diagnostic tool for AD faces challenges due to its susceptibility to noise and the complexity involved in the analysis. **Methods:** This study introduces a novel methodology employing three distinct stages for data-driven AD diagnosis: signal pre-processing, frame-level classification, and subject-level classification. At the frame level, convolutional neural networks (CNNs) are employed to extract features from spectrograms, scalograms, and Hilbert spectra. These features undergo fusion and are then fed into another CNN for feature selection and subsequent frame-level classification. After each frame for a subject is classified, a procedure is devised to determine if the subject has AD or not. **Results:** The proposed model demonstrates commendable performance, achieving over 80% accuracy, 82.5% sensitivity, and 81.3% specificity in distinguishing AD patients from healthy individuals at the subject level. **Conclusions:** This performance enables early and accurate diagnosis with significant clinical implications, offering substantial benefits over the existing methods through reduced misdiagnosis rates and improved patient outcomes, potentially revolutionizing AD screening and diagnostic practices. However, the model’s efficacy diminishes when presented with data from frontotemporal dementia (FTD) patients, emphasizing the need for further model refinement to address the intricate nuances associated with the simultaneous detection of various neurodegenerative disorders alongside AD.

## 1. Introduction

Dementia is a progressive neurocognitive syndrome affecting millions worldwide, with its prevalence expected to rise dramatically from 55 million cases in 2019 to 139 million by 2050 [1]. Among the most common subtypes of dementia are Alzheimer’s disease (AD) and frontotemporal dementia (FTD). AD is characterized by widespread neurodegeneration, particularly impacting memory and cognitive function, while FTD predominantly affects the frontal and anterior temporal lobes, leading to behavioral and language deficits [2]. Due to overlapping symptoms, such as apathy, agitation, and language difficulties, differentiating AD and FTD remains a significant clinical challenge [3]. This diagnostic complexity underscores the need for robust and accurate tools to distinguish between these disorders. Several methods diagnose dementia, particularly AD. While magnetic resonance imaging (MRI) and PET scans are effective, they are expensive and complex [4]. In contrast, electroencephalography (EEG) is a cost-effective, portable, and non-invasive alternative recommended by the European Federation of Neurological Societies Task Force guidelines in 2010 [5]. EEG measures brain activity through scalp electrodes, categorizing signals into delta, theta, alpha, beta, and gamma bands, which vary in frequency from low to high. The EEG signals are analyzed using fast Fourier transform (FFT) and Welch power spectrum density (PSD), with further decomposition by discrete wavelet transform (DWT) and empirical mode decomposition (EMD) [6,7,8,9]. Additionally, time–frequency techniques such as spectrograms and continuous wavelet transform (CWT) are utilized to analyze frequency changes over time [10]. However, EEG analysis remains challenging because of noise, non-linearity, and the difficulty in extracting specific information [11].

Due to the complexities of EEG analysis, many machine learning algorithms have been proposed for dementia detection [12]. Traditional machine learning methods, such as support vector machines (SVMs), k-nearest neighbors (k-NNs), logistic regression, linear discriminant analysis, decision trees, and random forests, are effective for dementia classification [13,14,15,16,17,18]. However, these methods rely on well-defined features, which can be a disadvantage if important attributes are missed or incorrectly identified. Recently, deep learning algorithms have gained popularity since they can learn and extract features from data without needing hand-crafted features or prior knowledge. For example, in [19], EEG signals were processed using a variational autoencoder to improve prediction accuracy. For EEG classification in dementia diagnosis, multi-layer perceptrons (MLPs) and convolutional neural networks (CNNs) are frequently used [20,21]. Models that handle sequence input, such as long short-term memory (LSTM) networks, are also beneficial for capturing temporal features [22,23].

Recent advances in dementia detection have explored diverse methodological approaches across multiple data modalities. In neuroimaging, Shen et al. [24] demonstrated the importance of patch selection strategies in 3D CNN-based AD classification using MRI data, achieving 89.6% accuracy with optimally sized cubic patches (48 × 48 × 48), highlighting how spatial sampling strategies significantly impact model performance. Complementing neuroimaging approaches, biochemical markers have shown promise for differential diagnosis, with Saraceno et al. [25] identifying serum Beta-Secretase 1 (BACE1) activity as a potential biomarker that distinguishes AD from FTD, showing significantly elevated levels in AD patients compared to both FTD patients and healthy controls. In the genomic domain, Abdelwahab et al. [26] achieved exceptional classification accuracies of 96.60% and 97.08% using PCA–CNN and SVD–CNN models, respectively, on gene expression data, demonstrating the effectiveness of combining dimensionality reduction with deep learning for AD prediction. For EEG-based approaches, Bajaj and Requena Carrión [27] introduced Spatio-Spectral Feature Images (SSFIs) to visualize and interpret CNN representations from multi-channel EEG signals, revealing distinct frequency band contributions and spatial patterns that enhance the understanding of deep learning model behavior in EEG analysis.

In addition, combinations of deep learning models have been employed. For example, in [28], features extracted from EEG data using bidirectional long short-term memory (BiLSTM) and a CNN were combined and fed into another neural network for Alzheimer’s disease classification. In the same research, an autoencoder was also utilized for data augmentation to further improve classification accuracy. Despite the numerous methods available for dementia classification, detecting Alzheimer’s disease remains challenging, especially in the presence of individuals with frontotemporal dementia (FTD).

In this study, we propose a novel methodology to extract features from a spectrogram, scalogram, and Hilbert spectrum using a CNN. The obtained features are then fused and fed into another CNN for feature selection and ultimate classification. The proposed model is evaluated by applying it on an EEG dataset that contained three groups, including AD, FTD, and cognitive normal (CN) patients, in order to discover its general characteristics regarding various dementia types. The proposed model introduces a novel approach by integrating three distinct signal representations—spectrograms, Hilbert spectra, and scalograms—within a unified EEG classification framework. This combination allows for capturing a broader range of time–frequency characteristics, offering a more comprehensive analysis of EEG data compared to traditional methods. A CNN is then used to extract hidden features from each representation, ensuring the detection of both time-varying and frequency-specific patterns. By fusing these features, the model enhances robustness and improves its classification performance and generalization across subjects. Despite the decent performance in detecting AD in healthy patients, with more than 80% accuracy, sensitivity, and specificity, the model did not maintain its effectiveness with the additional presence of FTD data.

The structure of the paper is as follows: Section 2 outlines the data acquisition process, while Section 3 presents the proposed methodology. Section 4 provides a summary of the experimental results, followed by a discussion in Section 5. Finally, the paper concludes in Section 6.

## 2. Data Acquisition

### 2.1. Participants

The EEG data used for evaluation in this study comprised the recordings from 88 participants acquired from the 2nd Department of Neurology of AHEPA General University Hospital of Thessaloniki [29]. All participants were diagnosed and categorized into three groups: Alzheimer’s disease (AD group) labeled as ‘A’, frontotemporal dementia (FTD group) labeled as ‘F’, and healthy control or cognitive normal (CN group) labeled as ‘C’. AD group included 36 participants whose average age was 66.4 ± 7.9, while FTD group consisted of 23 patients with the average age of 63.6 ± 8.2, and CN group comprised 29 participants with the average age of 67.9 ± 5.4. The disease duration was measured in months, with a median value of 25 months and an interquartile range (IQR) of 4.5 months, meaning that the middle 50% of participants had a disease duration between 24 and 28.5 months. Concerning the Alzheimer’s disease groups, no dementia-related comorbidities have been reported.

Additionally, the cognitive and neuropsychological state of each participant was evaluated by the international Mini-Mental State Examination (MMSE), with scores from 0 to 30, indicating severity of cognitive decline from high to low. The average MMSE for the AD group was 17.75 ± 4.5, for the FTD group 22.17 ± 2.64, and for the CN group 30.

### 2.2. EEG Recordings

EEG data were collected using a Nihon Kohden EEG 2100 clinical device, equipped with 19 scalp electrodes (Fp1, Fp2, F7, F3, Fz, F4, F8, T3, C3, Cz, C4, T4, T5, P3, Pz, P4, T6, O1, and O2) in accordance with the 10–20 international system and 2 reference electrodes (A1 and A2) placed on the mastoids for impedance checks and as reference electrodes. Each recording was conducted following the clinical protocol, with participants seated and their eyes closed. Prior to the initiation of each recording, skin impedance was ensured to be below 5 kΩ. The sampling rate was set to 500 Hz, with a resolution of 10 µV/mm. The duration of each recording was approximately 13.5 min for the AD group (range: 5.1–21.3 min), 12 min for the FTD group (range: 7.9–16.9 min), and 13.8 min for the CN group (range: 12.5–16.5 min).

The EEG recordings were pre-processed using a Butterworth band-pass filter with a frequency range of 0.5 to 45 Hz and subsequently re-referenced to the average value of A1 and A2. The signals were further refined using the Artifact Subspace Reconstruction (ASR) routine, an automatic artifact rejection technique that removes persistent or large-amplitude artifacts. This routine excluded data segments that exceeded a conservative threshold of a maximum window standard deviation of 17 within a 0.5 s window. Following this, the Independent Component Analysis (ICA) method in EEGLAB was employed to transform the 19 EEG signals into 19 ICA components. Components identified as ‘eye artifacts’ or ‘jaw artifacts’ by EEGLAB’s automatic classification method ‘ICLabel’ were then excluded.

## 3. Methodology

### 3.1. Model Overview

Our patient diagnosis model has three stages: signal pre-processing, frame-level classification, and subject-level classification. In the signal processing stage, continuous EEG recordings are segmented into 4 s frames with 50% overlap. The selection of a 4 s window length was based on several critical considerations: (1) neurophysiological requirements—this duration captures sufficient brain activity for meaningful time–frequency analysis, particularly for lower frequency bands (delta: 0.5–4 Hz and theta: 4–8 Hz) that are clinically relevant in dementia and require adequate temporal sampling for accurate representation; (2) computational efficiency—longer windows would exponentially increase processing time and memory requirements, while shorter windows may introduce spectral artifacts; (3) established neurological protocols—this duration aligns with standard EEG analysis practices in neurodegenerative disease research, where 2–8 s epochs are commonly employed; and (4) signal-to-noise ratio optimization—pilot studies demonstrated that 4 s segments provide optimal balance between temporal resolution and spectral accuracy. Each EEG channel from each frame is converted into three time–frequency representations: spectrogram, scalogram, and Hilbert spectrum. Each time–frequency representation of each channel is then converted into a (128 × 128) grayscale image to create a uniform feature structure across the spectrogram, scalogram, and Hilbert spectrum. Since there are 19 channels, each time–frequency representation results in 3-dimensional data with the size of (128 × 128 × 19) for each 4 s frame. In the frame-level classification stage, the 3-dimensional inputs from spectrogram, scalogram, and Hilbert spectrum are independently fed into a CNN, denoted by $CNN_{1}$, for feature extraction. Specifically, $CNN_{1-1}$ is the $CNN_{1}$ with spectrogram input, $CNN_{1-2}$ is the $CNN_{1}$ with scalogram input, and $CNN_{1-3}$ is the $CNN_{1}$ with Hilbert spectrum input. The final fully connected layer of $CNN_{1}$, with 50 neurons, produces a feature vector of size 50 for each time–frequency representation. These three feature vectors, derived from $CNN_{1-1}$, $CNN_{1-2}$, and $CNN_{1-3}$, are combined into a 2-dimensional input (3 × 50) and fed into another CNN, denoted by $CNN_{2}$ for frame classification. The final decision on whether a subject has AD is made using a modified majority voting system, where a threshold of 50% is set by default during training. This means that, if half of the frames are classified as AD, the subject is considered to have AD. The overall procedures are shown in Figure 1.

### 3.2. CNN Architecture Design and Training Parameters

#### 3.2.1. Architecture Rationale

The CNN architectures were specifically designed to optimize feature extraction from EEG time–frequency representations while maintaining computational efficiency. $CNN_{1}$ consists of four (3 × 3) convolutional layers with 32, 64, 128, and 256 filters, respectively, each followed by batch normalization and max-pooling layers (2 × 2 kernels, stride 2). This progressive increase in filter numbers follows established principles in computer vision, allowing the network to capture increasingly complex patterns: initial layers detect basic edges and textures representative of fundamental frequency components, while deeper layers identify high-level features corresponding to complex neural oscillation patterns characteristic of different dementia subtypes.

Max-pooling layers were chosen over average pooling to preserve dominant frequency features while reducing spatial dimensions as peak frequency responses are more diagnostically relevant than averaged responses in EEG analysis. The choice of 3 × 3 convolutional kernels balances receptive field coverage with parameter efficiency, enabling detection of local time–frequency patterns without excessive computational overhead.

$CNN_{2}$ employs a simpler architecture with only two convolutional layers (64 and 128 filters) as it operates on pre-extracted feature vectors rather than raw time–frequency images. This reduced complexity prevents overfitting while maintaining sufficient representational capacity for final classification decisions.

#### 3.2.2. Training Parameters and Optimization

All CNN models were trained using Adam optimizer with an initial learning rate of η = 0.001, $\beta_{1}$ = 0.9, and $\beta_{2}$ = 0.999. The learning rate was reduced by a factor of 0.5 when validation loss plateaued for 10 consecutive epochs, with a minimum learning rate of 1 ×$10^{-6}$. Training was conducted for a maximum of 150 epochs, with early stopping implemented when validation loss ceased to improve for 20 consecutive epochs to prevent overfitting.

Categorical cross-entropy loss function was employed for multi-class classification (AD, FTD, and CN), defined as$$L=-\sum_{i=1}^{N}\sum_{c=1}^{C}y_{i,c}\log\left({\hat{y}}_{i,c}\right)$$
where *N* is the number of samples, C is the number of classes, $y_{i,c}$ is the true label, and ${\hat{y}}_{i,c}$ is the predicted probability.

L2 regularization with λ = 0.0001 was applied to all convolutional and dense layers to prevent overfitting. Batch size was set to 32 to balance gradient stability with memory constraints. Data augmentation techniques, including random rotation (±15°), scaling (0.9–1.1), and Gaussian noise addition (σ = 0.01), were applied during training to increase dataset diversity and improve generalization capability.

### 3.3. Data Analysis

In this study, each EEG frame collected from each electrode was converted into a time–frequency representation. Thus, the original EEG frame was represented as 3-dimensional data, including information with respect to time, frequency, and channels. As aforementioned, there were 3 individual methods applied in parallel for data analysis. Those are spectrogram, scalogram, and Hilbert-spectrum.

#### 3.3.1. Spectrogram

A spectrogram, also known as the short-time Fourier transform (STFT), is a time–frequency representation of a signal. It is calculated using the squared magnitude of the Fourier transform derived from a short-time-segmented signal. It is used to analyze how the frequency content of a non-stationary signal changes over time. The process involves dividing the original signal x(n) of length $N_{x}$ into short segments using a sliding window g(n) of length *M* with a hopping step *R*. The number of segments *k* is given by(1)$$k=\left\lfloor \frac{N_{x}-\left(M-R\right)}{R}\right\rfloor$$
The spectrogram STFT X(f) is the concatenation of the discrete Fourier transforms (DFTs) of these segments:(2)$$X\left(f\right)=[X_{1}\left(f\right),X_{2}\left(f\right),\ldots ,X_{m}\left(f\right),\ldots ,X_{k}\left(f\right)]$$
where the DFT of segment *m* is(3)$$X_{m}\left(f\right)=\sum_{n=-\infty }^{\infty }x\left(n\right)g\left(n-mR\right)e^{-j2\pi fn},1\leq m\leq k$$
In this work, the sliding time window of 1 second with a hop size of 0.024 seconds results in 126 segments. The FFT was computed with 1024 points, covering frequencies from 0 to 250 Hz, which yields 92 data points for the 0.45 Hz to 45 Hz range. To minimize spectral leakage, a Hamming window function was used during segmentation. Consequently, the spectrogram for a 4-second frame of each channel, considering the frequency range from 0.45 Hz to 45 Hz, is represented as a matrix of size (92 × 126).

#### 3.3.2. Scalogram

Another method of converting EEG signals into a time–frequency representation applied in this study is continuous wavelet transform (CWT), also known as a scalogram. This wavelet transform decomposes the desired signal into a set of frequency components and displays the distribution in both the temporal and spectral domains by compressing, scaling, and shifting the signals. The scalogram is the 2D representation showing the energy density associated with a frequency at a certain time from the CWT. In order to perform CWT, the original signal (x(t)) is convoluted with a scaled CWT function (ψ(t)).(4)$$X_{cwt}\left(\beta ,\gamma \right)=\int_{-\infty }^{\infty }x\left(t\right)\psi_{\beta ,\gamma }^{*}\left(t\right)dt$$
where $X_{cwt}\left(\beta ,\gamma \right)$ is the CWT result, while $\psi_{\beta ,\gamma }^{*}\left(t\right)$ is a complex conjugate of scaled and shifted version of the mother wavelet ψ(t); β and γ are the scaling and shifting parameters. In this study, the scaling parameter is set based on a logarithmic scale with 21 voices per octave, ensuring smooth frequency resolution across multiple scales. The smallest scale $s_{0}$ is determined by the highest frequency of interest, approximately 45 Hz, leading to an initial scale of approximately 0.022 s. The scales are computed using the formula(5)$$s_{j}=s_{0}\cdot 2^{j/10},\;j=0,1,2,\ldots ,121$$
which covers frequencies from 45 Hz down to 0.45 Hz. The shifting parameter represents the time translation of the wavelet across the signal, enabling localized frequency analysis. The wavelet is shifted at each time step, corresponding to a sampling interval of 0.002 s determined by the 500 Hz sampling rate. This ensures precise temporal resolution while maintaining a balance between time and frequency localization. Consequently, the scalogram of a 4 s frame of each channel, considering the frequency range from 0.45 Hz to 45 Hz, has the size of (122 × 2000). The mother wavelet ψ(t) used in this work is Morse wavelet, whose Fourier transform is determined by Equation (Equation 6), where $a_{\beta ,\gamma }$ is a normalizing constant, and U(ω) is the unit step function.(6)$$\psi_{\beta ,\gamma }\left(\omega \right)=U\left(\omega \right)a_{\beta ,\gamma }\omega^{\beta }e^{-\omega^{\gamma }}$$

#### 3.3.3. Hilbert Spectrum

The Hilbert spectrum is another technique for performing time–frequency representation of non-stationary and non-linear signals. Compared to spectrograms, which only exploit the magnitude of the frequency with respect to the change in time, Hilbert spectrum depicts much detailed information at arbitrary time–frequency scales because it requires several computations, which are calculating intrinsic mode functions (IMFs) using empirical mode decomposition (EMD) and determining the instantaneous energy and instantaneous frequency of each IMF. In the first step, original signal *x* is decomposed into several components $c_{i}$ called IMFs and a residual *r* such that the original signal is the sum of all IMFs and the residual, as in Equation (Equation 7). The IMFs of higher-order *i* generally have larger energy in lower frequency bands. As long as the IMFs are determined, each IMF is then converted into analytic signal using Hilbert transform $\hat{H}\left\{c_{i}\right\}$. The analytic signal is able to be expressed in complex form as in Equation (Equation 8), whose real part is the original IMF, while the imaginary part is a version of the original real sequence with a 90° phase shift. Based on the complex form, the instantaneous amplitude $A_{i}$ and phase $\varphi_{i}$ of IMF of *i* order are defined, and the instantaneous angular frequency $\omega_{i}$ is determined by Equation (Equation 9).(7)$$x=\sum_{i}c_{i}+r$$(8)$$z_{i}=c_{i}+j\hat{H}\left\{c_{i}\right\}=A_{i}e^{j\varphi_{i}}$$(9)$$\omega_{i}=2\pi f_{i}=\frac{d\varphi_{i}}{dt}$$
where $f_{i}$ is the instantaneous frequency. From Equations (Equation 7)–(Equation 9), the original signal can be expressed as in Equation (Equation 10)(10)$$x=Real\sum_{i}A_{i}e^{j\;\int 2\pi f_{i}dt}+r$$

Finally, the Hilbert spectrum H(f,t) is able to be visualized by plotting the instantaneous amplitude *A* at the instantaneous frequency *f* as a function of time *t*. In this work, the computed Hilbert spectrum of a 4 s frame signal in one channel is a matrix with size of (128 × 2000), where 128 signifies the number of frequency points ranging from 0.45 to 45 Hz with the step of 0.35 Hz, and 2000 denotes the number of time points.

### 3.4. Convolutional Layer

Given the high temporal resolution of a 4 s signal sampled at 500 Hz, the data size of the scalogram and Hilbert spectrum across all 19 channels becomes considerably large. To address this and ensure consistency across modalities, the spectrogram, scalogram, and Hilbert spectrum for each channel are converted to grayscale images and resized to 128 × 128 pixels. This step reduces the data volume while standardizing the input format for CNN processing. Accordingly, each time–frequency representation across the 19 channels results in a final dimension of 128 × 128 × 19. The spectrogram and scalogram capture signal components across a range of frequencies, while the Hilbert spectrum further incorporates phase information, offering complementary insights into the underlying signal dynamics. This implies that the characteristics of signal were reflected by intra-component patterns rather than inter-component behaviors, which could be extracted using $CNN_{1}$. This extracting CNN independently extracts features from the spectrogram, scalogram, and Hilbert spectrum. The structure of $CNN_{1}$ was defined by the number of convolutional layers, dense layers, and hidden neurons in each dense layer. Specifically, as illustrated in Figure 2a, it included four (3 × 3) convolutional layers with batch normalization and max-pooling for each, a dense layer with 10 hidden neurons, and a 50-neuron fully connected layer used for further analysis. Features extracted from the spectrogram, scalogram, and Hilbert spectrum were then fused and fed into $CNN_{2}$ for ultimate feature selection. This selecting CNN had a similar structure to $CNN_{1}$ but included only two convolutional layers, as shown in Figure 2b. $CNN_{2}$ helped to obtain the optimal attributes of each feature extracted by $CNN_{1}$, and its output layer applied softmax activation to represent a categorical probability distribution of the desired EEG data frame.

### 3.5. Feature Similarity Analysis

The spectrogram depicts the power magnitude of the signal’s frequency components, while the scalogram captures transient patterns within the signal. In contrast, the Hilbert spectrum emphasizes phase changes over time. Consequently, these three time–frequency representations provide distinct information, and features extracted by $CNN_{1-1}$, $CNN_{1-2}$, and $CNN_{1-3}$ yield unique attributes. To ensure that the 50-neuron feature vectors do not exhibit similar characteristics, correlation coefficients are computed to analyze the linear dependence between each pair of features. The Pearson correlation coefficient for two variables $F_{a}$ and $F_{b}$ is determined by Equation (Equation 11): (11)$$\rho \left(F_{a},F_{b}\right)=\frac{cov(F_{a},F_{b})}{\sigma_{F_{a}}\sigma_{F_{b}}}$$
where $cov(F_{a},F_{b})$ is the covariance of $F_{a}$ and $F_{b}$, while $\sigma_{F_{a}}$ and $\sigma_{F_{b}}$ are the standard deviations of $F_{a}$ and $F_{b}$, respectively. The correlation coefficient $\rho (F_{a},F_{b})$ is defined as the cosine of the angle between two vectors $F_{a}$ and $F_{b}$ [30], which indicates the strength and direction of their linear relationship, ranging from −1 to 1. A value of ρ=1 indicates a perfect positive correlation. Conversely, ρ=−1 signifies a perfect negative correlation. A correlation of ρ=0 suggests no linear relationship between the variables. Values between 0 and 1 indicate varying degrees of positive correlation, with higher values representing stronger relationships. Similarly, values between −1 and 0 indicate negative correlations, where more negative values imply stronger inverse relationships.

For similarity analysis of $CNN_{1-1}$ feature, $CNN_{1-2}$ feature, and $CNN_{1-3}$ feature, we calculate the correlation matrix *R*, which includes the correlation coefficients for each pairwise feature combination, as shown in Equation (Equation 12): (12)$$R=\left[\begin{array}{ccc} 1 & \rho (CNN_{1-1},CNN_{1-2}) & \rho (CNN_{1-1},CNN_{1-3})\\ \rho (CNN_{1-2},CNN_{1-1}) & 1 & \rho (CNN_{1-2},CNN_{1-3})\\ \rho (CNN_{1-3},CNN_{1-1}) & \rho (CNN_{1-3},CNN_{1-2}) & 1 \end{array}\right]$$

### 3.6. Post-Processing

In this study, after categorizing the segmented EEG frames, the ultimate patient classification was decided by the most common category classified most frequently in the frame-based classification. In other words, the patient-based classification *Y* is the mode of the labels of the frame-based classification $\hat{Y}$.

### 3.7. Model Evaluation

#### 3.7.1. Statistical Analysis and Significance Testing

To ensure robust evaluation of our methodology, comprehensive statistical analysis was implemented. Statistical significance of performance differences between various feature extraction methods was evaluated using paired *t*-tests (α = 0.05) for pairwise comparisons when data satisfied normality assumptions (assessed using Shapiro–Wilk test); otherwise, Wilcoxon signed-rank tests were employed. For multiple group comparisons across different CNN architectures and feature combinations, one-way repeated measures ANOVA was conducted, followed by post hoc Tukey’s HSD tests when ANOVA revealed significant differences (*p* < 0.05).

Bootstrap resampling with n = 1000 iterations was performed to calculate 95% confidence intervals for all performance metrics (accuracy, sensitivity, specificity, and F1-score). Cohen’s d was computed to assess effect sizes for pairwise comparisons, with values ≥0.2, ≥0.5, and ≥0.8 considered small, medium, and large effects, respectively. Bonferroni correction was applied for multiple comparisons to control the family-wise error rate. All statistical analyses were conducted using Python (version 3.13.1) scipy.stats and statsmodels packages.

#### 3.7.2. Statistical Significance Analysis of Feature Extraction Methods

To evaluate the statistical significance of performance differences between the three feature extraction methods (spectrogram: $CNN_{1-1}$, scalogram: $CNN_{1-2}$, and Hilbert spectrum: $CNN_{1-3}$), paired *t*-tests were conducted for all pairwise comparisons across different classification tasks and validation approaches.

The paired *t*-test analysis revealed no statistically significant differences between any of the feature extraction methods in the AD–CN classification task using 5-fold cross-validation (α = 0.05). Specifically: $CNN_{1-1}$ vs. $CNN_{1-2}$ (*p* = 0.5178), $CNN_{1-2}$ vs. $CNN_{1-3}$ (*p* = 0.4514), and $CNN_{1-1}$ vs. $CNN_{1-3}$ (*p* = 0.8529). Similar non-significant results were observed for FTD–CN classification ($CNN_{1-1}$ vs. $CNN_{1-2}$: *p* = 0.6541; $CNN_{1-2}$ vs. $CNN_{1-3}$: *p* = 0.8884; $CNN_{1-1}$ vs. $CNN_{1-3}$: *p* = 0.69) and disease–healthy classification ($CNN_{1-1}$ vs. $CNN_{1-2}$: *p* = 0.1032; $CNN_{1-2}$ vs. $CNN_{1-3}$: *p* = 0.7830; $CNN_{1-1}$ vs. $CNN_{1-3}$: *p* = 0.0778).

The LOSO validation revealed one statistically significant difference in AD–CN classification between $CNN_{1-1}$ and $CNN_{1-3}$ (*p* = 0.0044), while other comparisons remained non-significant ($CNN_{1-1}$ vs. $CNN_{1-2}$: *p* = 0.0641; $CNN_{1-2}$ vs. $CNN_{1-3}$: *p* = 0.1036). This suggests that spectrogram-based and Hilbert-spectrum-based features show significantly different performance in the more stringent LOSO validation.

The statistical analysis reveals that, under standard 5-fold cross-validation, all three feature extraction methods demonstrate statistically equivalent performance across all classification tasks, supporting the rationale for the multimodal fusion approach where combining all three representations provides enhanced diagnostic capability through complementary information rather than relying on any single superior method.

#### 3.7.3. Confidence Intervals and Error Reporting

All performance metrics are reported with 95% confidence intervals calculated through bootstrap resampling. Figures displaying performance comparisons include error bars representing standard error of the mean across cross-validation folds. For box plots, the median, first and third quartiles, and outliers (beyond 1.5 × IQR) are clearly indicated to provide comprehensive distributional information.

Our proposed methodology encompasses distinct steps of feature extraction and feature selection. To gauge the effectiveness of each feature, we conducted a comparative analysis of the classification outcomes between the extracting CNN ($CNN_{1}$) and the selecting CNN ($CNN_{2}$). Additionally, we established a benchmark by comparing these outcomes with the results obtained using random forests applied to relative band power (RBP) of all 19 channels (RBP–RF), as highlighted for its superior performance in the study by [29]. The model underwent initial experimentation for AD–CN classification and disease (AD and FTD) and healthy (CN) classification for comprehensive analysis. Performance evaluation was carried out through 5-fold cross-validation, with 80% of subjects allocated for training and 20% for testing, considering accuracy, sensitivity, and specificity. The detailed analysis procedures are explicitly outlined in Figure 3.

To optimize our model, we used 5-fold cross-validation for hyperparameter tuning, which helped us to assess and adjust the model’s performance. We then verified the model using leave-one-subject-out (LOSO) cross-validation. LOSO is ideal for verification because it uses almost all the data in each iteration, providing a realistic test of the model’s ability to generalize to new subjects. By excluding one subject at a time, LOSO thoroughly evaluates the model’s performance and accounts for individual differences. While 5-fold cross-validation reduced the risk of overfitting during tuning, LOSO provided a stringent test, ensuring a comprehensive assessment. Using both methods enhances the reliability of our results and provides a better understanding of the model’s performance across different datasets.

In LOSO cross-validation, all segmented frames of one subject were excluded for testing, while the remaining subjects constituted the training set, within which data from 80% of the participants were used for training and data from the remaining 20% of the participants were used for validation. The trained model then classified all segments in the testing set, and majority voting was utilized for subject classification, which was subsequently compared to actual subject labels for subject-level evaluation. For frame-based evaluation, the classified and actual frame labels of each omitted subject were sequentially concatenated, resulting in a comprehensive confusion matrix for calculating performance metrics as described in Figure 4.

Finally, an evaluation was conducted to determine the significance of each EEG channel in identifying critical brain regions for detecting each class. This procedure, depicted in Figure 5, involved isolating the signals from individual EEG channels by inputting the spectrogram, scalogram, and Hilbert spectrum derived from each single channel into the proposed model to compare their performances across all channels. The model’s effectiveness for multi-class classification was assessed based on the average sensitivity across 5 folds of cross-validation, ensuring that models using different single channels were subjected to identical conditions, trained on the same training set, and tested on the same testing set in each repetition of the 5-fold cross-validation process. We opted not to employ LOSO due to its time-consuming nature. Channels demonstrating heightened sensitivity were identified as pivotal, whereas those with low sensitivity were deemed less crucial. The channel importance, ranging from 0 to 1, was then calculated by normalizing sensitivity results across all channels to rank their significance, with 0 indicating the least important or the channel with the lowest sensitivity and 1 indicating the most important channel with the highest sensitivity. Additionally, we utilized the EEGLAB function to visualize the sensitivity performances from all channels on a brain topographic map, facilitating the identification of significant brain regions essential for distinguishing between different classes.

## 4. Results

### 4.1. Model Classification Results

In the experiment designed to differentiate between AD and CN subjects using 5-fold cross-validation, Figure 6 illustrates the AD–CN classification performance of the proposed model, which incorporates individual time–frequency features such as STFT, CWT, HHT, and their fusion compared to the RBP–RF baseline method. The fusion model achieves the highest accuracy, sensitivity, and specificity, all exceeding 80%, outperforming each individual feature and the baseline. Among the individual features, all consistently surpass the RBP–RF baseline, with HHT demonstrating the best performance across all the metrics, followed by CWT and STFT. Additionally, Figure 7 presents the correlation between feature vectors extracted from each time–frequency attribute in the training, validation, and testing datasets. The results indicate that these three feature vectors exhibit low similarities, as evidenced by the maximum magnitude of the correlation coefficient being 0.068. This confirms that the fusion model does not suffer from overfitting.

However, the performance of the model substantially declines in the presence of FTD patients, as evidenced by the results in Figure 8, which illustrates that, while the model maintains high sensitivity in the disease–healthy classification, the specificity falls below 70%, indicating that the inclusion of FTD patients presents substantial challenges for the model. Furthermore, the three feature vectors do not exhibit strong similarity, confirmed by all the correlation coefficient magnitudes being below 0.45, as shown in Figure 9.

The evaluation results obtained from the 5-fold cross-validation demonstrated the proficiency of the proposed model in distinguishing between AD patients and CN subjects. To validate our approach, we repeated the AD–CN classification experiment using LOSO cross-validation. The results showed that combining all the features substantially improved the performance, achieving a sensitivity of 77.43% and a specificity of 70.14% at the frame level, as recorded in Table 1. Single features did not reach 70% specificity. Additionally, the model with feature fusion exceeded 80% in accuracy, sensitivity, specificity, and F1-score at the subject level, as reported in Table 2, with at least a 9% increase in specificity compared to single features. These findings demonstrate the robustness and effectiveness of our method in distinguishing between AD and CN subjects. Among the individual features, CWT has the highest performance, with 83.08% accuracy in subject-based AD–CN classification, followed by HHT, with an accuracy of 78.46%, and STFT, with an accuracy of 72.31%. Again, the feature vectors extracted by $CNN_{1-1}$, $CNN_{1-2}$, and $CNN_{1-3}$ exhibited low similarity, as indicated by the correlation coefficient magnitudes, which are smaller than 0.2, as shown in Figure 10.

### 4.2. Brain Region Importance

In order to analyze the significance of channels in class differentiation, each epoch from every channel underwent analysis in the proposed model. The resulting sensitivity scores for each class were compared across channels and arranged from highest to lowest, as depicted in Figure 11a. The sensitivity scores of all the channels were standardized to a scale of 0 to 1, representing the importance of each channel, with 0 indicating the least significance and 1 indicating the most crucial, as illustrated in Figure 11b.

According to the topographic maps presented in Figure 11, AD was accurately detected by nearly all the channels across various brain regions. Notably, electrodes F3 and F8 were identified as the least critical channels for detection in this context, as highlighted in Figure 11. In contrast, while FTD posed challenges for accurate identification, achieving a maximum average sensitivity of 22.86% using the most significant channel, Fp1, as illustrated in Figure 11, the evaluation of channel importance emphasized the frontal lobe, particularly channels Fp1, Fp2, F3, F7, and F8, as the most informative region compared to the other brain areas.

### 4.3. Comparison to Prior Studies

Table 3 presents a comparative analysis between our proposed model and recent state-of-the-art approaches in Alzheimer’s disease (AD) detection. While certain studies showcase remarkable performances with accuracy surpassing 90%, it is crucial to underscore that these achievements are based on datasets with a restricted number of participants with approximately 20 subjects in each group, supported by references such as [31], and [7]. However, as the dataset size expands to more than 30 patients in each group, the classification task tends to decline, as highlighted in [32] with 85% accuracy, [33] with 81% accuracy, and [34] with 67.84% accuracy. Many previous studies conducted classification at the frame level or segment base, which can lead to unreliable evaluation outcomes. This arises from variations where some patients have more accurately predicted segmented frames, while the majority of the segments from other patients are misclassified. Consequently, achieving high performance in frame-based classification does not necessarily guarantee conclusive results. It could be attributed to a scenario where a few patients’ segments are correctly predicted while the majority of other patients’ segmented frames are inaccurately predicted. Consequently, the model’s proficiency appears significant only for a few patients. Moreover, introducing an additional class such as FTD or mild cognitive impairment (MCI) into the dataset exacerbates the challenge of the classification task. This is evidenced by the results reported in [32,35], which achieved a precision of 70% for three-class classification, and in [34], which attained an accuracy of 63.12% for FTD–CN classification. Despite the commendable classification outcomes reported in certain studies, AD detection remains a persistent challenge due to the limited availability of recorded data, particularly concerning FTD—an aspect that has not been thoroughly explored in recent research endeavors.

## 5. Discussion

This study introduces a novel multimodal multi-stage deep learning model for diagnosing AD using EEG measurements. The proposed methodology integrates signal pre-processing, frame-level classification, and subject-level classification, leveraging CNNs to extract and combine features from STFT, CWT, and HHT. Based on the findings in Section 4, the combined use of these three features is both robust and effective, outperforming single features in distinguishing AD from CN patients. This fusion achieves over 80% accuracy in subject-level classification, as shown in Figure 6 and Table 2, and is comparable to or exceeds several recent state-of-the-art approaches with larger datasets, as reported in Table 3.

Among the individual features, HHT demonstrated superior performance in distinguishing AD from CN, achieving the highest accuracy, sensitivity, and specificity through 5-fold cross-validation, as depicted in Figure 6. Conversely, STFT showed the weakest performance, yielding the lowest results across the metrics. Through LOSO cross-validation, CWT emerged as the most effective feature for AD detection, with the highest sensitivity in both frame-level and subject-based classifications, as shown in Table 1 and Table 2. The limited contribution of STFT was further evident when fused with other features as there was no notable improvement in performance. In detail, the fusion of CWT and HHT achieves 74.01% accuracy, whereas the fusion of all three features results in 74.13% accuracy in frame-based classification, but there is no improvement in subject-based classification. This suggests that, while more frames are correctly classified, they are still within the same subjects, leading to no enhancement in subject-level accuracy. Overall, the results from both the 5-fold and LOSO validations confirm that CWT and HHT are informative features, while STFT is the least significant.

The inclusion of FTD data posed significant challenges to model performance, as evidenced by a drop in specificity below 70% in Figure 8 compared to the 80% specificity achieved for pure AD detection in Figure 6. While the model maintains high sensitivity in disease–healthy classification, it shows a tendency to misclassify healthy individuals as having cognitive diseases when FTD data are included. This implies difficulty in distinguishing FTD from CN as opposed to AD. The topographic maps in Figure 11 further elucidate this challenge: FTD affects localized brain regions, primarily in the frontal areas, including channels Fp1, Fp2, and F7, whereas the AD-associated patterns are more widespread, explaining why AD is easier to distinguish from CN than FTD. Additionally, frontal regions such as Fp1 and Fp2 are prone to artifacts, including eye blinks and muscular activity, complicating the extraction of clean EEG signals and contributing to the misclassification of healthy individuals as diseased [36]. These findings underscore the distinct neural patterns of AD and FTD, where AD exhibits more broadly distributed signatures, while FTD relies on localized frontal regions for classification. This limitation emphasizes the need for further refinement of the model to enhance its performance in the presence of FTD. Nevertheless, the analysis of topographic maps offers critical insights into the differential contributions of brain regions, providing valuable guidance for developing targeted diagnostic approaches, including the identification of critical EEG channels to enhance EEG-based diagnostic tools tailored to the neuropathological characteristics of each condition.

While our study demonstrates robust performance with over 80% accuracy in AD detection, certain limitations should be acknowledged for a comprehensive evaluation. The current dataset, although sufficient for validating our multimodal approach, represents a specific clinical population that may benefit from expansion to include more diverse demographic groups in future studies. This could further enhance the generalizability of our findings across different populations. The cross-sectional design provides strong initial validation, although longitudinal studies could offer additional insights into the model’s performance across disease progression stages. The 19-channel EEG montage, while clinically practical and cost-effective, represents a balance between diagnostic capability and accessibility, making our approach suitable for widespread clinical implementation.

From a clinical application perspective, our EEG-based diagnostic approach offers significant advantages for integration into existing healthcare workflows. The 30 min recording time aligns well with standard clinical appointments, while the non-invasive nature and cost-effectiveness of EEG make it particularly attractive for screening applications in both specialized memory clinics and primary care settings. In resource-limited environments where advanced neuroimaging may not be readily available, this approach could provide valuable diagnostic support for early dementia detection. However, successful clinical adoption will require the consideration of regulatory approval pathways, staff training for EEG acquisition and interpretation, and integration with existing electronic health record systems. The demonstrated performance superiority of our multimodal fusion approach over individual feature methods provides a strong foundation for clinical translation, particularly given the statistical validation showing complementary information from different time–frequency representations.

## 6. Conclusions

This study presents a novel multimodal multi-stage deep learning framework for diagnosing AD using EEG measurements. The proposed approach incorporates signal pre-processing, frame-level classification, and subject-level classification, utilizing CNNs to extract and integrate features from spectrograms, scalograms, and Hilbert spectra. The model demonstrates robust performance, achieving over 80% accuracy in distinguishing AD patients from cognitively normal individuals and delivering results that are comparable to or exceed those of recent state-of-the-art approaches.

While the model performs well in detecting AD, its effectiveness is challenged by the inclusion of FTD data. The reduced specificity observed when FTD data are incorporated highlights the difficulty in distinguishing FTD from cognitively normal states. This challenge stems from the localized nature of the FTD-affected brain regions, particularly in frontal areas prone to artifacts such as eye blinks and muscular activity. These findings emphasize the need to refine the model to improve its robustness and ability to generalize across neurodegenerative conditions.

To advance this research toward clinical implementation, several key validation steps are recommended. First, multi-center studies across diverse populations and clinical settings should validate the model’s generalizability and establish optimal implementation protocols. Second, longitudinal studies are needed to evaluate the model’s performance in tracking disease progression and early-stage detection capabilities. Third, targeted research should address the FTD classification challenges through specialized algorithms and artifact-resistant methods for the frontal regions. Finally, clinical workflow integration studies should assess practical implementation requirements, staff training needs, and cost-effectiveness compared to the current diagnostic approaches. We call upon the research community to advance EEG-based dementia diagnosis through systematic validation studies and clinical translation efforts, particularly given its potential to improve diagnostic accessibility in resource-limited settings.

Future research should aim to enhance the capacity of the model to differentiate between AD, FTD, and other neurodegenerative disorders. Identifying critical EEG channels associated with disease-specific neural patterns could further support the development of targeted diagnostic tools, advancing EEG-based diagnoses and enabling more accurate and timely detection, ultimately improving patient outcomes.

## Figures and Tables

**Figure 1 neurolint-17-00091-f001:**
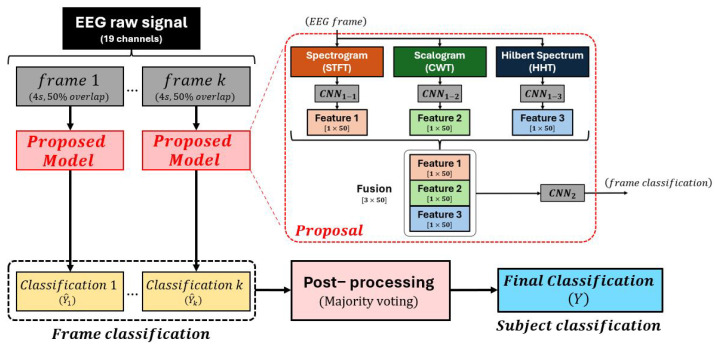
The proposed model.

**Figure 2 neurolint-17-00091-f002:**
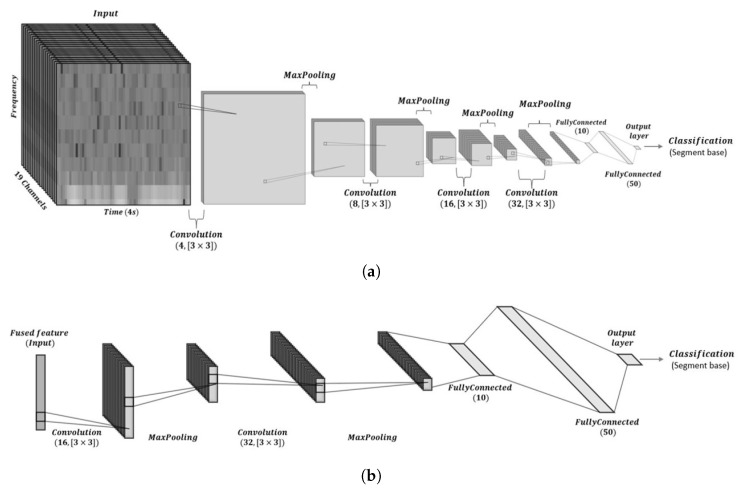
The architectures of the proposed model. (**a**) The architecture of the feature extractor ($CNN_{1}$). (**b**) The architecture of the feature selector ($CNN_{2}$).

**Figure 3 neurolint-17-00091-f003:**
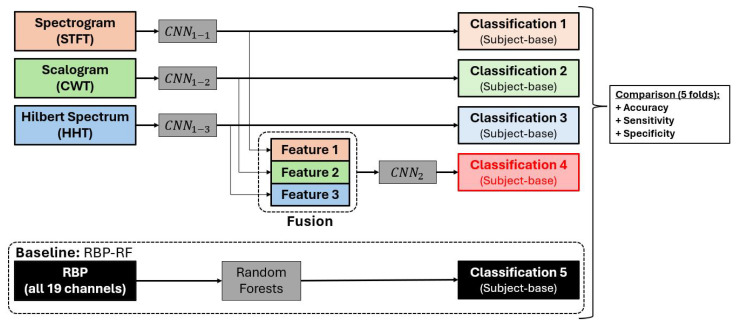
Procedure for performance evaluation of the proposed model against the baseline method.

**Figure 4 neurolint-17-00091-f004:**
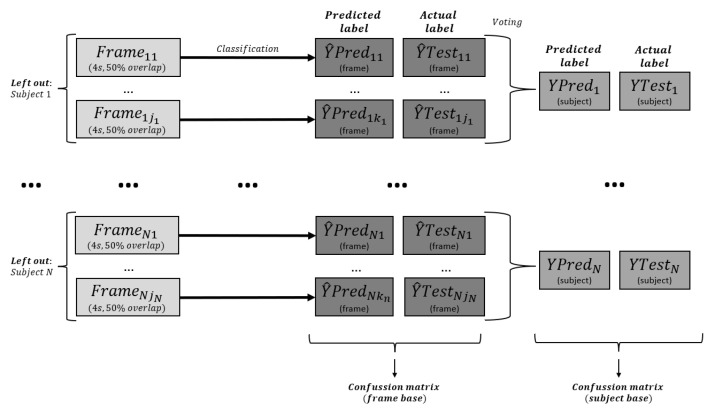
Confusion matrix analysis in leave-one-subject-out cross-validation.

**Figure 5 neurolint-17-00091-f005:**
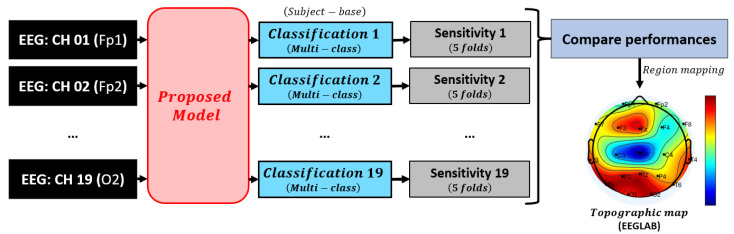
Evaluation of EEG channel importance using single-channel classification.

**Figure 6 neurolint-17-00091-f006:**
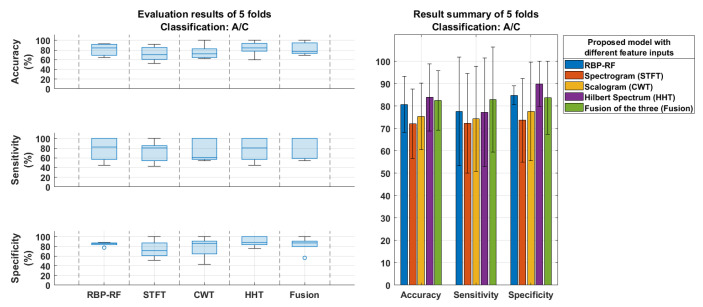
AD–CN classification performance of the proposed model using individual time–frequency features: STFT, CWT, HHT, and their fusion compared with the relative band power analyzed through random forest (RBP–RF).

**Figure 7 neurolint-17-00091-f007:**
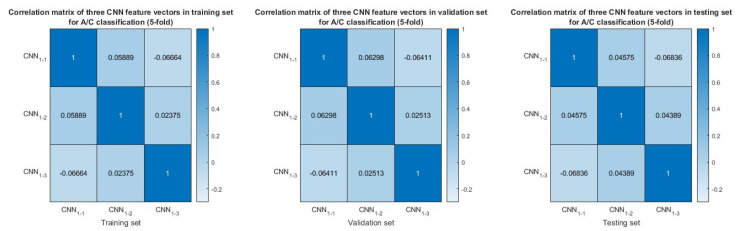
Similarity of CNN feature vectors in training set, validation set, and testing set for AD−CN classification (5−fold cross−validation) based on correlation matrices.

**Figure 8 neurolint-17-00091-f008:**
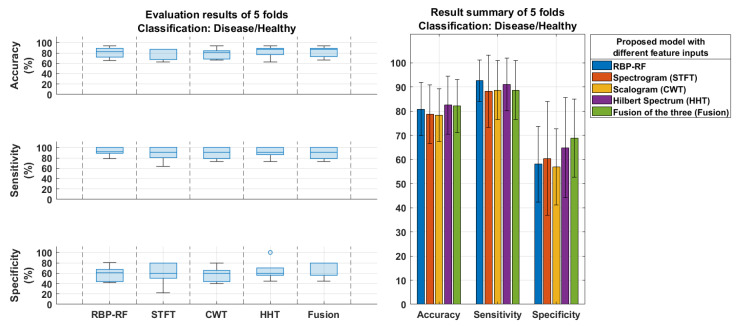
Disease (AD and FTD) and healthy (CN) classification performance of the proposed model using individual time–frequency features: STFT, CWT, HHT, and their fusion compared with the relative band power analyzed through random forest (RBP–RF).

**Figure 9 neurolint-17-00091-f009:**
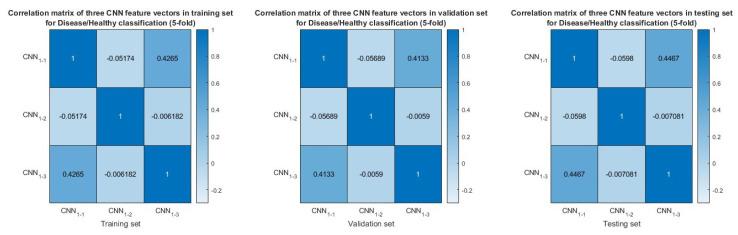
Similarity of CNN feature vectors in training set, validation set, and testing set for disease (AD and FTD) and healthy (CN) classification (5−fold cross−validation) based on correlation matrices.

**Figure 10 neurolint-17-00091-f010:**
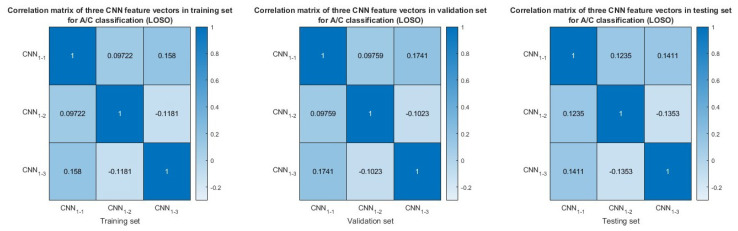
Similarity of CNN feature vectors in training set, validation set, and testing set for AD−CN classification (LOSO cross−validation) based on correlation matrices.

**Figure 11 neurolint-17-00091-f011:**
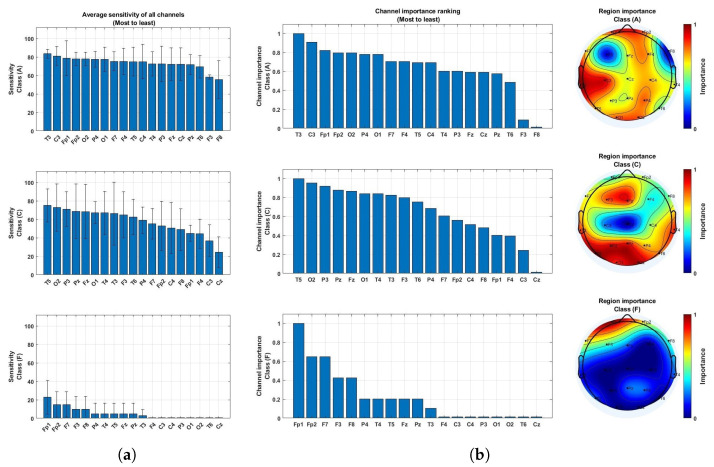
Importance ranking of EEG channels based on classification performance. (**a**) Performance of individual channels sorted from highest to lowest average sensitivity. (**b**) Importance ranking derived from average sensitivity.

**Table 1 neurolint-17-00091-t001:** Performance comparison of frame-based AD–CN classification (LOSO).

Feature input	Deep Learning Model	Accuracy (Frame-Based)	Sensitivity (Frame-Based)	Specificity (Frame-Based)	F1-Score (Frame-Based)
STFT	$CNN_{1-1}$	65.55%	67.65%	63.01%	68.24%
CWT	$CNN_{1-2}$	72.61%	76.02%	68.48%	75.23%
HHT	$CNN_{1-3}$	71.29%	79.23%	61.71%	75.13%
STFT + CWT	$CNN_{1-1}+CNN_{1-2}+CNN_{2}$	71.05%	73.59%	67.98%	73.56%
STFT + HHT	$CNN_{1-1}+CNN_{1-3}+CNN_{2}$	68.45%	70.49%	65.98%	70.98%
CWT + HHT	$CNN_{1-2}+CNN_{1-3}+CNN_{2}$	74.01%	77.24%	70.10%	76.48%
STFT + CWT + HHT	$CNN_{1-1}+CNN_{1-2}+CNN_{1-3}+CNN_{2}$	74.13%	77.43%	70.14%	76.61%

**Table 2 neurolint-17-00091-t002:** Performance comparison of subject-based AD–CN classification (LOSO).

Feature input	Deep Learning Model	Accuracy (Subject-Based)	Sensitivity (Subject-Based)	Specificity (Subject-Based)	F1-Score (Subject-Based)
STFT	$CNN_{1-1}$	72.31%	75.00%	68.97%	75.00%
CWT	$CNN_{1-2}$	83.08%	88.89%	75.86%	85.33%
HHT	$CNN_{1-3}$	78.46%	88.89%	65.52%	82.05%
STFT + CWT	$CNN_{1-1}+CNN_{1-2}+CNN_{2}$	78.46%	88.89%	65.52%	82.05%
STFT + HHT	$CNN_{1-1}+CNN_{1-3}+CNN_{2}$	81.54%	83.33%	79.31%	83.33%
CWT + HHT	$CNN_{1-2}+CNN_{1-3}+CNN_{2}$	84.62%	86.11%	82.76%	86.11%
STFT + CWT + HHT	$CNN_{1-1}+CNN_{1-2}+CNN_{1-3}+CNN_{2}$	84.62%	86.11%	82.76%	86.11%

**Table 3 neurolint-17-00091-t003:** Benchmarks of other recent related studies and this work.

Study	Dataset (Participants)	Feature Input	Model	Results	Frame/Subject Classification
Safi et al. [33]	30 AD 35 CN	Entropy Hjort parameters	SVM	Accuracy = 81% Sensitivity = 69.8% Specificity = 83.5%	Frame
Oltu et al. [31]	16 MCI 8 AD 11 CN	DWT Coherence	Bagged trees	Accuracy = 96.5% Sensitivity = 96.21% Specificity = 97.96%	Frame
Fouladi et al. [35]	61 HC 56 MCI 63 AD	CWT	Convolutional Autoencoder	Precision = 70% Recall = 88.92% F1 = 77.84%	Frame
Goker et al. [7]	24 AD 24 HC	Welch PSD	BiLSTM	Recall = 98.6% Precision = 99% F1 = 98.8% Accurracy = 98.85%	Frame
Jiao et al. [32]	330 AD 246 HC	PSD Hjort metrics STFT Entropy	LDA	Recall = 84.7% Precision = 87% F1 = 85.8% Accuracy = 85.8%	Frame
Kim et al. [34]	36 AD 29 CN	Global Field Power (GFP)	Gated Recurrent Unit Autoencoder (GRU-AE)	Accuracy = 67.84% Sensitivity = 80.24% Specificity = 52.93% F1 = 73.17%	Frame
This work (2025)	36 AD 29 CN	STFT CWT HHT	CNN	Accurracy = 84.62% Sensitivity = 86.11% Specificity = 82.76% F1 = 86.11%	Subject

## Data Availability

The dataset can be accessed by searching for its identifier on the OpenNeuro website.

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
