# Peer review of "A Multimodal Multi-Stage Deep Learning Model for the Diagnosis of Alzheimer’s Disease Using EEG Measurements"

_2035-8377, 2025, doi:10.3390/neurolint17060091_

Round 1
Reviewer 1 Report
Comments and Suggestions for Authors
In this paper, a novel methodology was proposed that contains three distinct stages for data-driven Alzheimer's disease diagnosis. The proposed model contains a signal pre-processing stage, a frame-level classification stage, and a subject-level classification stage, respectively. At the frame level stage, Convolutional Neural Networks (CNN) were employed to extract features from the spectrogram, scalogram, and Hilbert spectrum. These features were then undergoing a fusion and fed into another CNN for feature selection and subsequent frame-level classification. After each frame for a subject was classified, a procedure was devised to determine if the subject had AD or not. Here are some major revisions;
- Why did the authors use the spectrogram, scalogram, and Hilbert spectrum images?
- Actually, the proposed model was not seen as novel. So, please indicate the novelties carefully.
- How did you decide on a 50% overlap for the frame level stage? Did you use other ratios?
- I think an ablation study is needed.
Author Response
Response to Reviewer 1 Comments
We sincerely thank the reviewer for their constructive feedback and insightful comments. We have carefully addressed each concern and made substantial revisions to improve the manuscript. Below are our detailed responses to each comment:
Comment 1: Why did the authors use the spectrogram, scalogram, and Hilbert spectrum images?
Response: We have added comprehensive justification for our choice of these three time-frequency representations in the enhanced introduction and methodology sections. The selection was based on their complementary characteristics that capture distinct aspects of EEG signal properties. The spectrogram (STFT) captures time-frequency energy distributions with fixed temporal and frequency resolution, providing stable representation of frequency content changes over time. The scalogram (CWT) offers adaptive time-frequency resolution through wavelet-based decomposition, with better temporal resolution for high frequencies and better frequency resolution for low frequencies. The Hilbert spectrum reveals instantaneous frequency characteristics and phase information through Empirical Mode Decomposition, capturing non-linear and non-stationary signal properties that are not visible in traditional Fourier-based methods. This combination allows comprehensive characterization of the complex neural dynamics underlying different dementia phenotypes, as each representation captures distinct aspects of EEG signal properties that are individually insufficient for optimal classification.
Comment 2: Actually, the proposed model was not seen as novel. So, please indicate the novelties carefully.
Response: We have extensively clarified the novel contributions of our work in the enhanced introduction to address this important concern. Our work presents several key novelties that distinguish it from previous research in EEG-based dementia classification. This represents the first systematic integration of three complementary time-frequency representations (spectrogram, scalogram, Hilbert spectrum) within a unified deep learning framework for dementia classification. We introduce a novel multimodal fusion architecture that combines CNN-based feature extraction from each representation with a secondary CNN for feature selection and classification. Our approach also provides a comprehensive differential diagnosis framework addressing the challenging clinical problem of distinguishing AD from FTD using EEG, which has been rarely explored in previous literature. Additionally, we implement a multi-stage classification framework with frame-level and subject-level decision making using modified majority voting. Unlike previous EEG-based studies that typically employ single transformations or limited combinations, our approach systematically harnesses the synergistic information from all three representations, enabling comprehensive characterization of neural dynamics that cannot be achieved through individual methods.
Comment 3: How did you decide on a 50% overlap for the frame level stage? Did you use other ratios?
Response: We have added detailed justification for the 50% overlap in the enhanced methodology section to address this methodological question. The 50% overlap was selected based on several important considerations that balance performance and computational efficiency. This represents standard practice in signal processing for ensuring adequate temporal continuity between adjacent segments, which is crucial for maintaining the integrity of time-varying neural patterns. The chosen overlap provides an optimal balance between data augmentation and computational efficiency, as higher overlap increases dataset size but also computational burden proportionally. Furthermore, this ratio ensures optimal information retention while avoiding excessive redundancy between consecutive frames that could lead to overfitting. Pilot studies conducted during our preliminary analysis demonstrated that 50% overlap provided optimal balance between classification performance and processing time compared to other ratios tested. We acknowledge that this represents a methodological choice that could benefit from systematic evaluation of different overlap ratios in future work, which we have noted as a recommendation for future research in our enhanced conclusion section.
Comment 4: I think an ablation study is needed.
Response: We respectfully note that our study includes comprehensive ablation studies that systematically examine the contribution of each component, though we have enhanced the presentation of these analyses based on the reviewer's feedback. Our existing ablation analysis includes individual feature performance evaluation where we systematically compared the performance of each time-frequency representation (CNN₁₋₁, CNN₁₋₂, CNN₁₋₃) individually across all classification tasks. We conducted feature combination analysis evaluating pairwise combinations, demonstrating that CWT+HHT achieved 74.01% accuracy compared to all three features achieving 74.13% accuracy, showing the incremental contribution of each component. Statistical significance testing through paired t-tests was conducted for all pairwise comparisons across different validation approaches, providing rigorous statistical validation of our findings. Both 5-fold and LOSO validation were performed to assess robustness and generalizability, while performance was systematically compared against Random Forest with Relative Band Power (RBP-RF) as a baseline. We have enhanced this analysis by adding detailed statistical results showing that individual methods perform equivalently under standard validation, supporting the multimodal fusion approach. Feature similarity analysis using correlation coefficients demonstrates that the three representations provide distinct, complementary information, while channel-wise importance analysis identifies critical brain regions for each classification task. The comprehensive evaluation demonstrates that the enhanced performance comes from the synergistic combination of complementary information rather than the superiority of any single method, validating our multimodal approach.

Reviewer 2 Report
Comments and Suggestions for Authors
Deep learning makes an important contribution to the medical field by providing highly successful approaches in the early diagnosis and diagnosis of Alzheimer's disease. In the present study, the authors utilise EEG data for AD diagnosis. In this context, the authors present a CNN-based deep learning approach. In the proposed study, over 80% in accuracy, sensitivity, and specificity results were obtained in AD disease diagnosis. The paper is generally well presented and can contribute to the literature. Within the scope of the study, enough references related to the study were presented.
Some suggestions to the authors within the scope of the study:
1. In the study, the authors should include more explanatory statements about why they use a CNN-based approach.
2. LSTM-based approaches can give better results for EEG data in the literature. For this reason, the authors should include a discussion section in the paper. They should address the superiority of their method.
3. In Table 3, the authors compare the results obtained in the paper with different data. It is normal for the results to be in this way because of the limited data and different data. However, it would be good to make methodological comparisons in the relevant section.
Author Response
Response to Reviewer 2 Comments
We sincerely thank the reviewer for their positive feedback and constructive suggestions. We are pleased that the reviewer found our paper well-presented and capable of contributing to the literature. We have carefully addressed each suggestion and made significant revisions to strengthen the manuscript. Below are our detailed responses:
General Comments Acknowledgment
We appreciate the reviewer's recognition of our study's contribution to the medical field through the application of deep learning for Alzheimer's disease diagnosis using EEG data. The achievement of over 80% accuracy, sensitivity, and specificity demonstrates the potential clinical value of our multimodal approach, and we are grateful for the positive assessment of our reference coverage and overall presentation.
Comment 1: Authors should include more explanatory statements about why they use a CNN-based approach.
Response: We have substantially enhanced the methodology section to provide comprehensive justification for our CNN-based approach. The selection of CNNs was motivated by several key advantages that make them particularly suitable for our multimodal time-frequency analysis framework. CNNs excel at extracting spatial patterns from image-like data, which is crucial since we convert our time-frequency representations (spectrograms, scalograms, and Hilbert spectra) into 128×128 grayscale images. The hierarchical feature learning capability of CNNs allows automatic extraction of increasingly complex patterns, starting from basic time-frequency features in early layers to high-level diagnostic patterns in deeper layers, eliminating the need for manual feature engineering. The translation invariance property of CNNs ensures robust detection of neural patterns regardless of their temporal or frequency location within the representation, which is essential for capturing the variable manifestations of dementia-related neural changes. Additionally, the shared parameter structure of CNNs provides computational efficiency while maintaining sufficient model capacity for learning complex relationships between different brain regions and frequency bands. We have added detailed architectural rationale explaining how the progressive increase in filter numbers (32→64→128→256) enables hierarchical pattern recognition, and why max-pooling layers preserve dominant frequency features while reducing spatial dimensions. The CNN architecture also facilitates our multimodal fusion strategy, as features extracted from each time-frequency representation can be effectively combined and processed by the secondary CNN for final classification decisions.
Comment 2: LSTM-based approaches can give better results for EEG data in the literature. For this reason, authors should include a discussion section addressing the superiority of their method.
Response: We have enhanced our discussion section to address this important comparison between CNN and LSTM approaches for EEG analysis. While LSTM-based approaches have indeed shown success in EEG analysis due to their ability to capture temporal dependencies in sequential data, our CNN-based approach offers distinct advantages for our specific multimodal time-frequency analysis framework. LSTMs are particularly effective when processing raw EEG time series data where temporal sequence relationships are crucial, but our approach transforms EEG signals into time-frequency image representations where spatial pattern recognition becomes more important than sequential temporal modeling. The time-frequency images inherently capture temporal information within their spatial structure, making CNN's spatial pattern recognition capabilities more suitable than LSTM's sequential processing. Furthermore, our multimodal fusion of three different time-frequency representations creates a comprehensive feature space that benefits more from CNN's ability to extract and combine spatial patterns than from LSTM's temporal sequence modeling. The computational efficiency of our CNN approach also enables practical processing of multiple time-frequency representations simultaneously, whereas LSTM-based processing of equivalent multimodal data would require significantly more computational resources. Additionally, our statistical analysis demonstrates that the complementary information from different time-frequency representations provides diagnostic value that cannot be easily captured by single-modal LSTM approaches processing raw EEG sequences. We acknowledge that LSTM approaches have their merits, particularly for applications focusing on temporal dynamics of raw EEG signals, but our CNN-based multimodal approach is specifically optimized for extracting diagnostic patterns from time-frequency image representations, which aligns better with our research objectives of comprehensive dementia classification using multiple complementary signal transformations.
Comment 3: In Table 3, authors compare results with different data. It is normal for results to be in this way because of limited data and different data. However, it would be good to make methodological comparisons in the relevant section.
Response: We have enhanced the results and discussion sections to provide more comprehensive methodological comparisons that go beyond simple numerical performance comparisons. While we acknowledge that direct performance comparisons across different datasets have inherent limitations due to data variability, participant demographics, and recording protocols, we have expanded our analysis to include detailed methodological comparisons that highlight the fundamental differences in approach. Our multimodal time-frequency analysis represents a distinct methodological advancement compared to existing single-modal approaches reported in the literature. Most previous studies focus on either spectral power analysis, single time-frequency transformations, or traditional machine learning with hand-crafted features, whereas our approach systematically integrates three complementary signal representations within a unified deep learning framework. We have added discussion of how our CNN-based feature extraction from image-converted time-frequency representations differs from approaches that apply deep learning directly to raw EEG signals or traditional frequency domain features. The methodological comparison now includes analysis of computational complexity, feature interpretability, and clinical applicability across different approaches. We also discuss how our multi-stage classification framework with frame-level and subject-level decisions provides advantages over single-stage classification methods commonly used in previous studies. Furthermore, we have expanded the comparison to include discussion of validation strategies, highlighting how our combination of 5-fold cross-validation and Leave-One-Subject-Out validation provides more rigorous evaluation than single validation approaches used in many comparison studies. The enhanced methodological comparison demonstrates that while numerical performance may vary across different datasets, our systematic multimodal approach offers conceptual and practical advantages that make it particularly suitable for clinical implementation in diverse healthcare settings.

Reviewer 3 Report
Comments and Suggestions for Authors
The manuscript holds significant potential, promising novelty and scientific robustness. However, it would greatly benefit from the comprehensive methodological clarifications, precise statistical validations, and enhanced clarity regarding practical clinical applications.
Abstract:
- For clarity in the abstract, specific numerical details, such as exact sensitivity and specificity values (e.g., “over 80%”), could be mentioned.
- The abstract should also explicitly indicate the potential clinical implications, such as early and accurate diagnosis, and the benefits over existing methods, like reduced misdiagnosis rates and improved patient outcomes.
Introduction:
- Clearly outlines the challenge in differentiating Alzheimer’s Disease (AD) from Frontotemporal Dementia (FTD) in the introduction.
- The introduction should also provide context regarding EEG as an accessible diagnostic tool. Furthermore, the novelty of the multimodal approach, incorporating spectrogram, scalogram, and Hilbert spectrum, should be explicitly stated and compared with previous EEG-based studies. This will help to pique the interest of our readers. We should also specify the research question or hypothesis to orient our readers effectively. In addition, it would be beneficial to specify the type of EEG used in the study, whether it's resting-state EEG, event-related potentials, or another type, to provide a clearer understanding of the methodology and to enhance the manuscript's robustness.
Methodology:
- The methodology needs to be clearer on the CNN training parameters (epochs, learning rate, loss functions, optimization techniques).
- Also, justifying why a 4-second window was chosen (as opposed to other window lengths) would strengthen methodological clarity in the methodology. This could be due to the need to capture sufficient brain activity for analysis while maintaining computational efficiency. Providing this rationale will enhance the manuscript's robustness and help the reader understand the methodological decisions made.
- The methodology should clarify why specific CNN architectures were selected, including the number of layers, type of pooling layers, etc. This will help the reader understand the rationale behind these choices and how they contribute to the model's effectiveness.
- Also, the methodology should provide statistical significance of differences between performance metrics (e.g., using paired t-tests or ANOVA), especially when comparing different feature extraction methods.
- Finally, confidence intervals or error bars in figures should be provided in the methodology to enhance interpretability.
Discussion:
- In the discussion section, it's crucial to thoroughly discuss the limitations of our current study, such as the limited diversity in the dataset and generalizability concerns. This comprehensive discussion will demonstrate our responsibility as researchers and ensure that our audience is fully informed and aware of the potential challenges.
- Also, expand discussion on practical clinical applications, how this method could be integrated into existing clinical workflows, and potential barriers to clinical adoption.
Conclusion:
- In the conclusion, it's important to include explicit recommendations or a call to action for future research or clinical validation studies. This will guide the reader on potential next steps and how the findings of this study can be further validated and applied in clinical practice.
Reference:
- While the references are adequately current and relevant, it would be beneficial to consider adding more recent references (2023-2025) to reflect the latest advancements in EEG-based dementia diagnosis. This will ensure that the authors and readers are up-to-date with the most recent research in the field.
Author Response
Response to Reviewer 3 Comments
We sincerely thank the reviewer for their comprehensive and detailed feedback. The thorough review has significantly contributed to improving the quality and clarity of our manuscript. We have carefully addressed each section-specific comment and made substantial revisions throughout the paper. Below are our detailed responses:
Abstract:
- For clarity in the abstract, specific numerical details, such as exact sensitivity and specificity values (e.g., “over 80%”), could be mentioned.
- The abstract should also explicitly indicate the potential clinical implications, such as early and accurate diagnosis, and the benefits over existing methods, like reduced misdiagnosis rates and improved patient outcomes.
Response: We have comprehensively enhanced the abstract to address both suggestions regarding numerical details and clinical implications. The revised abstract now includes specific numerical values stating "achieving over 80% accuracy, 82.5% sensitivity, and 81.3% specificity" instead of the vague "over 80%" reference. We have also explicitly incorporated clinical implications by adding detailed discussion of early and accurate diagnosis capabilities, reduced misdiagnosis rates, and improved patient outcomes compared to existing methods. The enhanced abstract now clearly states that "this performance enables early and accurate diagnosis with significant clinical implications, offering substantial benefits over existing methods through reduced misdiagnosis rates and improved patient outcomes, potentially revolutionizing AD screening and diagnostic practices." These additions provide readers with concrete performance metrics and clear understanding of the clinical value proposition of our approach.
Introduction
- Clearly outlines the challenge in differentiating Alzheimer’s Disease (AD) from Frontotemporal Dementia (FTD) in the introduction.
- The introduction should also provide context regarding EEG as an accessible diagnostic tool. Furthermore, the novelty of the multimodal approach, incorporating spectrogram, scalogram, and Hilbert spectrum, should be explicitly stated and compared with previous EEG-based studies. This will help to pique the interest of our readers. We should also specify the research question or hypothesis to orient our readers effectively. In addition, it would be beneficial to specify the type of EEG used in the study, whether it's resting-state EEG, event-related potentials, or another type, to provide a clearer understanding of the methodology and to enhance the manuscript's robustness.
Response: We have substantially expanded the introduction to address all the comprehensive suggestions provided. The enhanced introduction now clearly outlines the challenge in differentiating AD from FTD by adding detailed explanation of why distinguishing between these conditions is clinically challenging, including the consequences of misdiagnosis and the complexity of early-stage presentations. We have provided extensive context regarding EEG as an accessible diagnostic tool, emphasizing its advantages for various clinical settings, screening programs, and longitudinal monitoring, particularly in resource-limited environments. The novelty of our multimodal approach has been explicitly stated and compared with previous EEG-based studies, highlighting how most existing work uses single-modal or limited combinations while our approach represents the first systematic integration of three complementary time-frequency representations. We have added a clear research question and hypothesis stating "Can a multimodal deep learning approach that simultaneously leverages spectrograms, scalograms, and Hilbert spectra significantly improve the accuracy of EEG-based differential diagnosis between AD, FTD, and cognitively normal individuals?" The introduction now specifies that our study utilizes resting-state EEG recordings and explains why this approach is valuable for capturing intrinsic neural network alterations. Additionally, we have integrated recent research findings from 2023-2024 studies to provide comprehensive context of current advancements in the field.
Methodology
- The methodology needs to be clearer on the CNN training parameters (epochs, learning rate, loss functions, optimization techniques).
- Also, justifying why a 4-second window was chosen (as opposed to other window lengths) would strengthen methodological clarity in the methodology. This could be due to the need to capture sufficient brain activity for analysis while maintaining computational efficiency. Providing this rationale will enhance the manuscript's robustness and help the reader understand the methodological decisions made.
- The methodology should clarify why specific CNN architectures were selected, including the number of layers, type of pooling layers, etc. This will help the reader understand the rationale behind these choices and how they contribute to the model's effectiveness.
- Also, the methodology should provide statistical significance of differences between performance metrics (e.g., using paired t-tests or ANOVA), especially when comparing different feature extraction methods.
- Finally, confidence intervals or error bars in figures should be provided in the methodology to enhance interpretability.
Response: We have thoroughly enhanced the methodology section to address all technical clarity concerns raised. The CNN training parameters are now comprehensively detailed, including learning rate of 0.001 with Adam optimizer, batch size of 32, 150 epochs with early stopping after 20 consecutive epochs, categorical cross-entropy loss function with mathematical definition, L2 regularization with λ = 0.0001, and specific data augmentation techniques. We have added detailed justification for the 4-second window choice, explaining that this duration captures sufficient brain activity for meaningful time-frequency analysis particularly for lower frequency bands clinically relevant in dementia, balances computational efficiency, optimizes signal-to-noise ratio based on pilot studies, and aligns with established neurological protocols. The CNN architecture selection rationale has been extensively clarified, explaining the progressive filter increase (32→64→128→256) for hierarchical pattern recognition, the choice of max-pooling over average pooling for preserving dominant frequency features, and the 3×3 kernel selection for optimal receptive field coverage. We have added comprehensive statistical analysis including paired t-tests for pairwise comparisons, one-way ANOVA for multiple groups, bootstrap resampling for 95% confidence intervals, Cohen's d for effect size assessment, and specific results showing that individual feature extraction methods perform equivalently under standard validation, supporting our multimodal fusion approach. The methodology now includes detailed specifications for confidence intervals and error reporting in figures.
Discussion
- In the discussion section, it's crucial to thoroughly discuss the limitations of our current study, such as the limited diversity in the dataset and generalizability concerns. This comprehensive discussion will demonstrate our responsibility as researchers and ensure that our audience is fully informed and aware of the potential challenges.
- Also, expand discussion on practical clinical applications, how this method could be integrated into existing clinical workflows, and potential barriers to clinical adoption.
Response: We have enhanced the discussion section to provide balanced coverage of study limitations and clinical applications. The revised discussion acknowledges limitations in a constructive manner, noting that while our study demonstrates robust performance with over 80% accuracy, the current dataset represents a specific clinical population that could benefit from expansion to include more diverse demographic groups in future studies. We frame these as opportunities for enhancement rather than fundamental flaws, emphasizing that the 19-channel EEG montage represents a practical balance between diagnostic capability and accessibility for widespread clinical implementation. The discussion has been expanded to include detailed analysis of practical clinical applications, describing how the 30-minute recording time aligns with standard clinical appointments and how the cost-effectiveness of EEG makes it particularly attractive for screening in both specialized memory clinics and primary care settings. We discuss integration strategies for different healthcare environments, including resource-limited settings where advanced neuroimaging may not be available. The enhanced discussion addresses potential barriers to clinical adoption including regulatory approval pathways, staff training requirements, and integration with existing electronic health record systems, while emphasizing the demonstrated advantages of our multimodal fusion approach.
Conclusion
In the conclusion, it's important to include explicit recommendations or a call to action for future research or clinical validation studies. This will guide the reader on potential next steps and how the findings of this study can be further validated and applied in clinical practice.
Response: We have enhanced the conclusion section to include explicit recommendations and calls to action for future research and clinical validation studies. The revised conclusion provides specific guidance including the need for multi-center studies across diverse populations and clinical settings to validate generalizability, longitudinal studies to evaluate disease progression tracking and early-stage detection capabilities, targeted research to address FTD classification challenges through specialized algorithms and artifact-resistant methods, and clinical workflow integration studies to assess practical implementation requirements and cost-effectiveness. We have included a direct call to action stating "We call upon the research community to advance EEG-based dementia diagnosis through systematic validation studies and clinical translation efforts, particularly given its potential to improve diagnostic accessibility in resource-limited settings." The enhanced conclusion provides readers with concrete next steps and actionable recommendations while maintaining focus on the clinical translation potential of our approach.
References
While the references are adequately current and relevant, it would be beneficial to consider adding more recent references (2023-2025) to reflect the latest advancements in EEG-based dementia diagnosis. This will ensure that the authors and readers are up-to-date with the most recent research in the field.
Response: We have updated our reference list to include more recent publications from 2023-2025 to reflect the latest advancements in EEG-based dementia diagnosis. The enhanced introduction now incorporates recent research findings including Shen et al. (2023) on MRI-based 3D CNN approaches, Saraceno et al. (2024) on biochemical markers for AD/FTD differentiation, Abdelwahab et al. (2023) on gene expression analysis with deep learning, and Bajaj and Requena Carrión (2023) on EEG visualization techniques. These additions ensure that our work is positioned within the most current research landscape and demonstrates awareness of the latest developments in multimodal approaches to dementia diagnosis across different data types.

Round 2
Reviewer 3 Report
Comments and Suggestions for Authors
The authors have comprehensively addressed all points raised in our review and thoroughly and clearly explained the changes implemented, significantly improving the manuscript’s clarity, methodological rigor, and clinical relevance.